# Diagnostic Performance in Differentiating COVID-19 from Other Viral Pneumonias on CT Imaging: Multi-Reader Analysis Compared with an Artificial Intelligence-Based Model

Francesco Rizzetto [1,2,*], Luca Berta [3], Giulia Zorzi [3,4], Antonino Cincotta [1,2], Francesca Travaglini [1], Diana Artioli [1], Silvia Nerini Molteni [5], Chiara Vismara [5], Francesco Scaglione [5,6], Alberto Torresin [3,7], Paola Enrica Colombo [3,7], Luca Alessandro Carbonaro [1,6] and Angelo Vanzulli [1,6]

1  Department of Radiology, ASST Grande Ospedale Metropolitano Niguarda, Piazza Ospedale Maggiore 3, 20162 Milan, Italy
2  Postgraduate School of Diagnostic and Interventional Radiology, Università degli Studi di Milano, Via Festa del Perdono 7, 20122 Milan, Italy
3  Department of Medical Physics, ASST Grande Ospedale Metropolitano Niguarda, Piazza Ospedale Maggiore 3, 20162 Milan, Italy
4  Postgraduate School of Medical Physics, Università degli Studi di Milano, Via Giovanni Celoria 16, 20133 Milan, Italy
5  Chemical-Clinical and Microbiological Analyses, ASST Grande Ospedale Metropolitano Niguarda, Piazza Ospedale Maggiore 3, 20162 Milan, Italy
6  Department of Oncology and Hemato-Oncology, Università degli Studi di Milano, Via Festa del Perdono 7, 20122 Milan, Italy
7  Department of Physics, Università degli Studi di Milano, Via Giovanni Celoria 16, 20133 Milan, Italy
*  Correspondence: francesco.rizzetto@unimi.it

**Abstract:** Growing evidence suggests that artificial intelligence tools could help radiologists in differentiating COVID-19 pneumonia from other types of viral (non-COVID-19) pneumonia. To test this hypothesis, an R-AI classifier capable of discriminating between COVID-19 and non-COVID-19 pneumonia was developed using CT chest scans of 1031 patients with positive swab for SARS-CoV-2 ($n = 647$) and other respiratory viruses ($n = 384$). The model was trained with 811 CT scans, while 220 CT scans ($n = 151$ COVID-19; $n = 69$ non-COVID-19) were used for independent validation. Four readers were enrolled to blindly evaluate the validation dataset using the CO-RADS score. A pandemic-like high suspicion scenario (CO-RADS 3 considered as COVID-19) and a low suspicion scenario (CO-RADS 3 considered as non-COVID-19) were simulated. Inter-reader agreement and performance metrics were calculated for human readers and R-AI classifier. The readers showed good agreement in assigning CO-RADS score (Gwet's AC2 = 0.71, $p < 0.001$). Considering human performance, accuracy = 78% and accuracy = 74% were obtained in the high and low suspicion scenarios, respectively, while the AI classifier achieved accuracy = 79% in distinguishing COVID-19 from non-COVID-19 pneumonia on the independent validation dataset. The R-AI classifier performance was equivalent or superior to human readers in all comparisons. Therefore, a R-AI classifier may support human readers in the difficult task of distinguishing COVID-19 from other types of viral pneumonia on CT imaging.

**Keywords:** COVID-19; artificial intelligence; radiomics; lung; tomography (X-ray computed)

## 1. Introduction

Coronavirus Disease 2019 (COVID-19) is a complex infectious disease caused by the Severe Acute Respiratory Syndrome Coronavirus 2 (SARS-CoV-2), which has caused more than half a billion cases and 6 million deaths since it was first reported in late 2019 [1].

From a radiological point of view, CT findings of SARS-CoV-2 pulmonary infection include ground-glass opacities, areas of crazy-paving pattern, and consolidations. Such

alterations are usually multiple and bilateral, with patchy distribution and predominant involvement of basal and subpleural lung regions [2,3]. However, not all COVID-19 patients exhibit these characteristics, making differential diagnosis from other pulmonary diseases challenging [4,5]. In particular, the typical appearance of COVID-19 partially overlaps with the CT findings of different types of viral pneumonia, similar to those from adenovirus and rhinovirus [6]. This reduces the specificity of chest CT and raises the risk of false-positive diagnosis, especially in case of low incidence and prevalence of COVID-19 [7].

To facilitate the evaluation of chest CT of patients with suspected lung involvement by SARS-CoV-2, the COVID-19 Reporting and Data System (CO-RADS) score was proposed [8]. This scheme provides a standardized five-point scale to express the suspicion of COVID-19 pneumonia on chest CT images, demonstrating excellent diagnostic performance and moderate-to-substantial interobserver agreement [9]. Nevertheless, the CO-RADS category 3, which accounts for equivocal findings, still implies positivity to COVID-19 in 20–40% of cases [8,10].

Given the need for efficient tools for the detection and differential diagnosis of COVID-19, there has been a considerable drive to develop solutions based on quantitative imaging, such as radiomics and artificial intelligence (AI) [11]. Many authors have pointed out the potential added value of AI models in differentiating COVID-19 from other types of pneumonia, with accuracy ranging from 80% to over 95% [12–15]. However, in most cases, the diagnostic performance of AI models was assessed by comparing COVID-19 with heterogenous pulmonary conditions, often including bacterial infections [15–17], whose distinct features can ease the classification task.

In this work, we designed a multi-reader study to assess the performance of a radiomics-based AI classifier in the radiological challenge of discriminating COVID-19 from other types of viral-only pneumonia with microbiologically established etiology. We also simulated two distinct suspicion scenarios to investigate the impact of the varying epidemiological conditions on diagnostic performance.

## 2. Materials and Methods

### 2.1. Study Design and Imaging Data

This study was retrospectively conducted in a single high-volume referral hospital for the management of the COVID-19 pandemic. The Local Ethics Committee (decision number 188-22042020) approved the study and waived informed consent since data were collected retrospectively and processed anonymously.

Chest CT scans of 1031 consecutive patients with a positive PCR nasopharyngeal swab for SARS-CoV-2 (COVID-19, *n* = 647) and other respiratory viruses (non-COVID-19, *n* = 384) were collected. The panel of non-COVID-19 viruses detected included: adenovirus, bocavirus 1/2/3/4, coronavirus 229E/NL63/OC43, enterovirus, influenza virus A/B viruses, metapneumovirus, parainfluenza virus 1/2/3/4, rhinovirus A/B/C, and respiratory syncytial virus A/B. Patients with evidence of bacterial coinfection in their clinical documentation were excluded.

The CT scans of COVID-19 patients were performed between March 2020 and April 2021, while CT scans of non-COVID-19 patients were performed between January 2015 and October 2019 (i.e., before SARS-CoV-2 started circulating). For both groups, the CT scans were acquired within 15 days of serological evidence of infection.

Chest CT examinations were performed with different CT scanners (Somatom Definition Edge—Siemens, Somatom Sensation 64—Siemens, Brilliance 64—Philips) and with the same patient set-up (supine position with arms over the head during a single breath-hold, in keeping with the patient compliance). The main acquisition parameters were: tube voltage = 80–140 kV; automatic tube current modulation; pitch = 1; matrix = 512 × 512. All acquisitions were reconstructed with high-resolution thorax kernels and a slice thickness of 3 mm.

## 2.2. Artificial Intelligence-Based Model

The collected CT images were used to develop a radiomic-based Neural Network (R-AI) classifier exploiting a Multi-Layer Perceptron architecture to discriminate between COVID-19 and non-COVID-19 pneumonia. In particular, the classifier was trained with 811 CT images ($n$ = 496 COVID-19, $n$ = 315), while the remaining 220 CT images ($n$ = 151 COVID-19, $n$ = 69 non-COVID-19) were used as an independent validation dataset, applying a threshold on the predicted values of 0.5. Details about the R-AI classifier, including development and tuning, were previously described [18].

The R-AI classifier provided as output the probability (0.00–1.00) that the analyzed CT scan belonged to a COVID-19 patient.

## 2.3. Reader Evaluation

Three radiologists with >10 years of experience (Readers 1–3) and one radiology resident with 3 years of experience (Reader 4), all employed at a high-volume COVID-19 referral hospital, were enrolled to evaluate the 220 CT scans of the independent validation dataset. The four readers were blinded to the original radiologic report and all non-imaging data, including the acquisition date of the CT scans. They were asked to assign each case the CO-RADS score [8] (1 to 5) to express the increasing suspicion of COVID-19. To properly simulate a realistic clinical scenario, the readers were instructed to interpret the CT findings, assuming that the patients had an acute condition (e.g., presentation at the Emergency Department).

Additionally, as an estimate of disease severity, for each patient, the readers visually assessed the extent of pulmonary involvement expressed as a percentage of the total lung volume, rounded to the nearest 10%.

The test was performed using a program developed in JavaScript that automatically opened to the reader the anonymized CT series in random order. After the reader had assigned the CO-RADS score through a dialog box, the program automatically loaded the CT of the next patient in random order.

## 2.4. Data Analysis

Continuous variables were reported as median values with 25th and 75th percentiles (Q1–Q3) of their distribution; categorical variables were expressed as counts and percentages, with the corresponding 95% confidence interval (95%CI) using the Wilson method [19].

The chance-corrected inter-reader agreement for the assigned CO-RADS score was tested using Gwet's second-order agreement coefficient (AC2) with ordinal weights [20]. AC2 was chosen to correct for the partial agreement occurring when comparing ordinal variables with multiple readers and because it is less affected by prevalence and marginal distribution [21–23]. The level of agreement was interpreted following Altman's guidelines [24]. Weighted percentage agreement was reported as well [25].

To account for equivocal results (i.e., CO-RADS 3), two different scenarios were simulated: a high suspicion scenario, where CO-RADS 3 results were considered as COVID-19 patients, and a low suspicion scenario, where CO-RADS 3 results were considered as non-COVID-19 patients together with CO-RADS 1 and 2.

Sensitivity (SE), specificity (SP), accuracy (ACC), positive likelihood ratio (PLR), and negative likelihood ratio (NLR) of human readers in discriminating COVID-19 patients from non-COVID-19 patients were calculated for both high and low suspicion scenarios. The same metrics of diagnostic performance were also calculated for the R-AI classifier.

Moreover, a further subanalysis was conducted to compare the performance of human readers and the R-AI classifier in challenging cases when two or more readers had assigned a CO-RADS 3 score.

Significant differences in the diagnostic performance of the readers and the R-AI classifier were tested using Cochran's Q test with a post-hoc pairwise McNemar test.

The data analysis was generated using the Real Statistics Resource Pack software (Release 6.8) (www.real-statistics.com (accessed on 1 October 2022)) for Microsoft Excel (Microsoft Corporation, Redmond, Washington, DC, USA) and GraphPad Prism 8.4.0 (GraphPad Software, La Jolla, CA, USA).

Statistical significance was established at the *p* < 0.050 level, applying Bonferroni's correction for multiple comparisons when appropriate.

## 3. Results

The demographic characteristics of the patient population are reported in Table 1. Specifically, the 220 patients of the independent validation set consisted of 159 (72%) males and 61 (28%) females and had a median age of 68 (Q1–Q3: 59–78) years. Averaging between the different readers, the median extent of their pulmonary disease was 33% (Q1–Q3: 20–53%) of the total lung volume. Median interval between CT scans and molecular swabs was of 1 (Q1–Q3: 0–2) days for COVID-19 and 3 (Q1–Q3: 1–6) days for non-COVID-19 patients.

**Table 1.** Demographic characteristics of the study population.

| | **Total Population** | | |
| | **All Patients** | **COVID-19** | **Non-COVID-19** |
| --- | --- | --- | --- |
| Age | 66 (55–77) | 67 (55–78) | 66 (54–74) |
| Sex | | | |
|     Male | 694 (67%) | 458 (71%) | 236 (61%) |
|     Female | 337 (33%) | 189 (29%) | 148 (39%) |
| Virus | | | |
|     SARS-CoV-2 | 647 (63%) | 647 (100%) | - |
|     Adenovirus | 14 (1%) | - | 14 (4%) |
|     Coronavirus 229E/NL63/OC43 | 29 (3%) | - | 29 (8%) |
|     Enterovirus | 5 (0%) | - | 5 (1%) |
|     Influenza virus A/B | 147 (14%) | - | 147 (38%) |
|     Bocavirus 1/2/3/4 | 12 (1%) | - | 12 (3%) |
|     Metapneumovirus | 22 (2%) | - | 22 (6%) |
|     Parainfluenza virus 1/2/3/4 | 25 (2%) | - | 25 (7%) |
|     Rhinovirus A/B/C | 94 (9%) | - | 94 (24%) |
|     Respiratory syncytial virus A/B | 36 (3%) | - | 36 (9%) |
| Total | 1031 (100%) | 647 (100%) | 384 (100%) |
| | **Train Set** | | |
| | **All Patients** | **COVID-19** | **Non-COVID-19** |
| Age | 66 (54–77) | 67 (55–78) | 65 (54–74) |
| Sex | | | |
|     Male | 535 (66%) | 347 (70%) | 188 (60%) |
|     Female | 276 (34%) | 149 (30%) | 127 (40%) |
| Virus | | | |
|     SARS-CoV-2 | 496 (61%) | 496 (100%) | - |
|     Adenovirus | 12 (1%) | - | 12 (4%) |
|     Coronavirus 229E/NL63/OC43 | 25 (3%) | - | 25 (8%) |
|     Enterovirus | 4 (0.5%) | - | 4 (1%) |
|     Influenza virus A/B | 119 (15%) | - | 119 (38%) |
|     Bocavirus 1/2/3/4 | 10 (1%) | - | 10 (3%) |
|     Metapneumovirus | 17 (2%) | - | 17 (5%) |
|     Parainfluenza virus 1/2/3/4 | 19 (2%) | - | 19 (6%) |
|     Rhinovirus A/B/C | 79 (10%) | - | 79 (25%) |
|     Respiratory syncytial virus A/B | 30 (4%) | - | 30 (10%) |
| Total | 811 (100%) | 496 (100%) | 315 (100%) |

**Table 1.** *Cont.*

| | Independent Validation Set | | |
| | **All Patients** | **COVID-19** | **Non-COVID-19** |
|---|---|---|---|
| Age | 68 (59–78) | 67 (59–79) | 68 (60–75) |
| Sex | | | |
|     Male | 159 (72%) | 111 (74%) | 48 (70%) |
|     Female | 61 (28%) | 40 (26%) | 21 (30%) |
| Virus | | | |
|     SARS-CoV-2 | 151 (69%) | 151 (100%) | - |
|     Adenovirus | 2 (1%) | - | 2 (3%) |
|     Coronavirus 229E/NL63/OC43 | 4 (2%) | - | 4 (6%) |
|     Enterovirus | 1 (0.005%) | - | 1 (1%) |
|     Influenza virus A/B | 28 (13%) | - | 28 (41%) |
|     Bocavirus 1/2/3/4 | 2 (1%) | - | 2 (3%) |
|     Metapneumovirus | 5 (2%) | - | 5 (7%) |
|     Parainfluenza virus 1/2/3/4 | 6 (3%) | - | 6 (9%) |
|     Rhinovirus A/B/C | 15 (7%) | - | 15 (22%) |
|     Respiratory syncytial virus A/B | 6 (3%) | - | 6 (9%) |
| Total | 220 (100%) | 151 (100%) | 69 (100%) |

CO-RADS scores assigned by each reader were detailed in Table 2. Considering the global performance of the four readers, the error rate was 17% (95%CI: 14–20%) in classifying patients as COVID-19 and 32% (95%CI: 27–38%) in classifying them as non-COVID-19. Notably, some discrepancies could be observed since Reader 3 tended to assign CO-RADS 2 score more frequently in both the COVID-19 and non-COVID-19 groups compared to the other readers. However, inter-reader agreement in assigning the CO-RADS score was good, with an ordinal-weighted AC2 of 0.71 (95%CI: 0.67–0.76; $p < 0.001$) and a weighted percentage agreement of 91% (95%CI: 90–92%; $p < 0.001$). Perfect agreement was obtained between the four readers in 53/220 (24%) cases, distributed as follows: 3/53 (6%) CO-RADS 1, 20/53 (38%) CO-RADS 2, and 30/53 (57%) CO-RADS 5.

**Table 2.** CO-RADS scores assigned to COVID-19 and non-COVID-19 patients by the four readers.

| CO-RADS | COVID-19 Patients | | | | | Non-COVID-19 Patients | | | | | Total Readings |
| | Reader 1 | Reader 2 | Reader 3 | Reader 4 | Total | Reader 1 | Reader 2 | Reader 3 | Reader 4 | Total | |
|---|---|---|---|---|---|---|---|---|---|---|---|
| 1 | 7 (5%) | 4 (3%) | 9 (6%) | 12 (8%) | 32 (5% | 16 (23%) | 10 (14%) | 11 (16%) | 18 (26%) | 55 (20%) | 87 (10%) |
| 2 | 25 (17%) | 7 (5%) | 29 (19%) | 8 (5%) | 69 (11%) | 28 (41%) | 23 (33%) | 48 (70%) | 27 (39%) | 126 (46%) | 195 (22%) |
| 3 | 21 (14%) | 25 (17%) | 24 (16%) | 20 (13%) | 91 (15%) | 16 (23%) | 25 (36%) | 6 (9%) | 15 (22%) | 62 (22%) | 152 (17%) |
| 4 | 36 (24%) | 33 (22%) | 46 (30%) | 21 (14%) | 135 (22%) | 4 (6%) | 6 (9%) | 3 (4%) | 0 (0%) | 13% (5%) | 149 (17%) |
| 5 | 62 (41%) | 82 (54%) | 43 (28%) | 90 (60%) | 277 (46%) | 5 (7%) | 5 (7%) | 1 (1%) | 9 (13%) | 20 (7%) | 297 (34%) |
| | 151 (100%) | | | | 604 (100%) | 69 (100%) | | | | 276 (100%) | 880 (100%) |

The rate of patients classified as CO-RADS 1 (normal/noninfectious) was 10% (95%CI: 8–12%), while the rate of CO-RADS 3 (equivocal cases) was 17% (95%CI: 15–20%). Specifically, 43 (20%) cases received a CO-RADS 3 score from two or more readers, of which 26 (60% of 43) were COVID-19 patients and 17 (40% of 43) were non-COVID-19 patients. On the other hand, the R-AI classifier misclassified 21% (95%CI: 15–28%) of the COVID-19 patients and 22% (95%CI: 14–33%) of the non-COVID-19 patients. Exemplary cases are shown in Figure 1.

Regarding the diagnostic performance in identifying COVID-19 pneumonia, full results are provided in Table 3, Figures 2 and 3. Considering all the readers, SE = 83% (95%CI: 80–86%), SP = 66% (95%CI: 60–71%), ACC = 78% (95%CI: 75–80%), PLR = 2.35 (95%CI: 2.00–2.76), and NLR = 0.25 (95%CI: 0.21–0.30) were observed in the high suspicion scenario. On the other hand, SE = 68% (95%CI: 64–72%), SP = 88% (95%CI: 84–92%), ACC = 75% (95%CI: 72–77%), PLR = 5.70 (95%CI: 4.12–7.89), and NLR = 0.36 (95%CI: 0.32–0.41) were obtained in the low suspicion scenario.

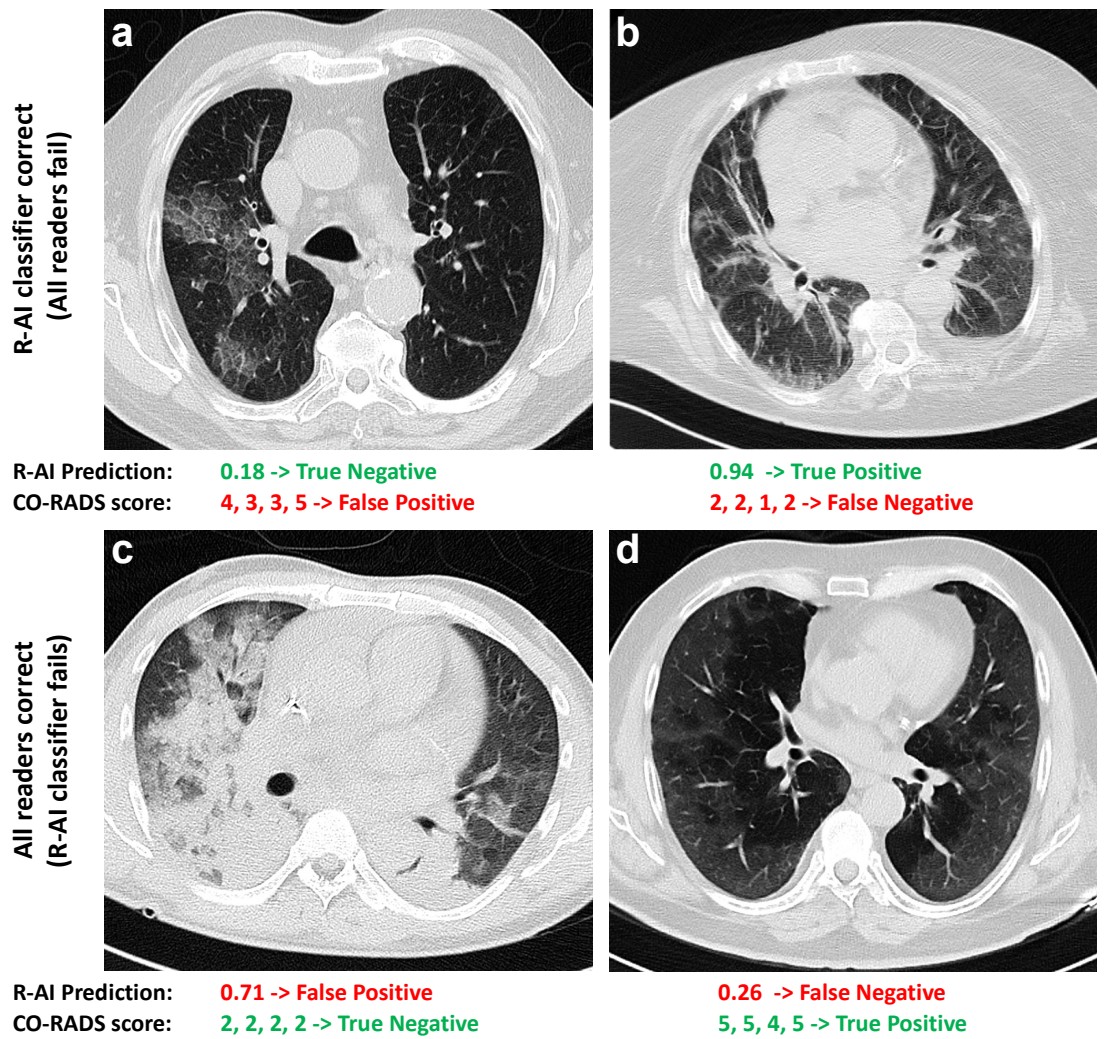

**Figure 1.** Exemplary cases of different situations occurred in the classification task. For each case, the probability output of the R-AI classifier and the CO-RADS scores assigned by the readers are reported. Final diagnoses by molecular testing were: (**a**) parainfluenza virus 4 pneumonia; (**b**) COVID-19 pneumonia; (**c**) rhinovirus pneumonia; (**d**) COVID-19 pneumonia.

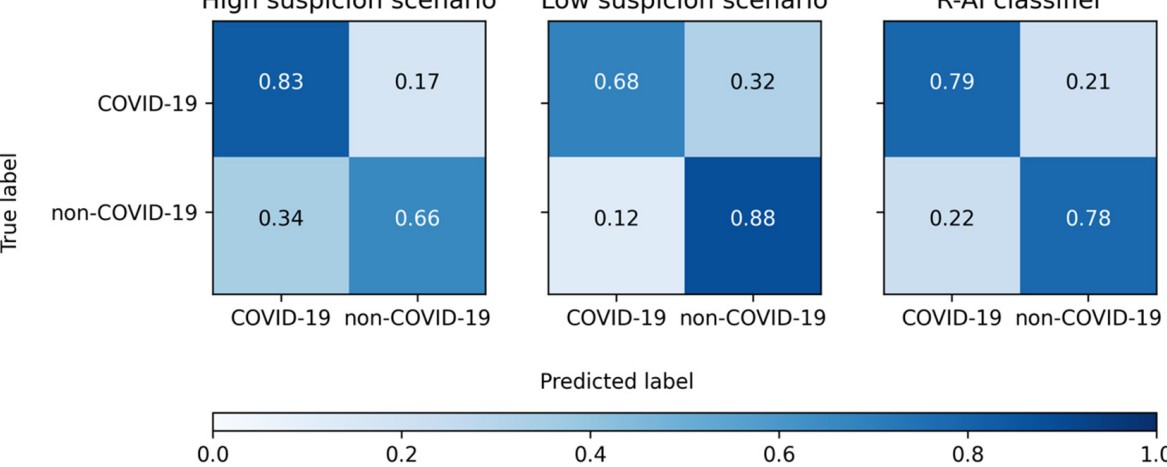

**Figure 2.** Confusion matrices of the global performance of the four readers in the high and low suspicion scenarios and the radiomic-based artificial intelligence (R-AI) classifier in distinguishing COVID-19 from other types of viral pneumonia.

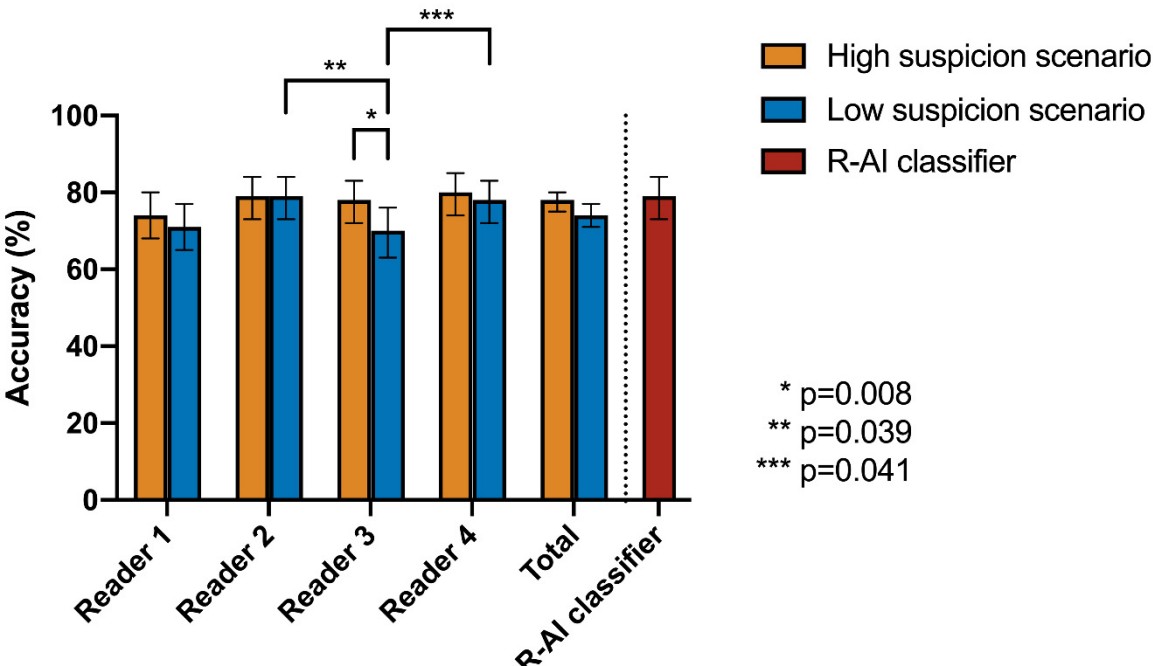

**Figure 3.** Comparison between the accuracy of the radiomic-based artificial intelligence (R-AI) classifier and the four readers in differentiating COVID-19 from other types of viral pneumonia in both high and low suspicion scenarios. Bars on graph boxes represent the 95% confidence interval of the accuracy values. Significant results of pairwise McNemar test after Bonferroni's correction were reported.

**Table 3.** Diagnostic performance of radiomic-based artificial intelligence classifier (R-AI) and human readers in classifying the patients in the high and low suspicion scenarios.

|  | SE | SP | ACC | PLR | NLR |
|---|---|---|---|---|---|
| | | | High Suspicion Scenario | | |
| Reader 1 | 79% (71–85%) | 64% (51–75%) | 74% (68–80%) | 2.18 (1.57–3.01) | 0.33 (0.23–0.47) |
| Reader 2 | 93% (87–96%) | 48% (36–60%) | 79% (73–84%) | 1.78 (1.41–2.24) | 0.15 (0.08–0.28) |
| Reader 3 | 75% (67–82%) | 86% (75–93%) | 78% (72–83%) | 5.16 (2.89–9.23) | 0.29 (0.22–0.39) |
| Reader 4 | 87% (80–92%) | 65% (53–76%) | 80% (74–85%) | 2.49 (1.79–3.47) | 0.20 (0.13–0.32) |
| Total | 83% (80–86%) | 66% (60–71%) | 78% (75–80%) | 2.42 (2.05–2.86) | 0.25 (0.21–0.31) |
| | | | Low suspicion scenario | | |
| Reader 1 | 65% (57–72%) | 87% (77–94%) | 72% (65–78%) | 4.98 (2.68–9.25) | 0.40 (0.32–0.51) |
| Reader 2 | 76% (69–83%) | 84% (73–92%) | 79% (73–84%) | 4.78 (2.76–8.27) | 0.28 (0.21–0.38) |
| Reader 3 | 59% (51–67%) | 94% (86–98%) | 70% (63–76%) | 10.17 (3.89–26.57) | 0.44 (0.36–0.53) |
| Reader 4 | 74% (66–80%) | 87% (77–94%) | 78% (72–83%) | 5.64 (3.04–10.44) | 0.30 (0.23–0.40) |
| Total | 68% (64–72%) | 88% (84–92%) | 75% (72–77%) | 5.70 (4.12–7.89) | 0.36 (0.32–0.41) |
| R-AI | 79% (71–85%) | 78% (67–87%) | 79% (73–84%) | 3.63 (2.30–5.72) | 0.27 (0.19–0.38) |

95% confidence intervals were reported in parentheses. SE: sensitivity; SP: specificity; ACC: accuracy; PLR: positive likelihood ratio; NLR: negative likelihood ratio.

When considering the R-AI classifier, it achieved SE = 79% (95%CI: 71–85%), SP = 78% (95%CI: 67–87%), ACC = 79% (95%CI: 73–84%), PLR = 3.63 (95%CI: 2.30–5.72), and NLR = 0.27 (95%CI: 0.19–0.38) in distinguishing COVID-19 from non-COVID-19 pneumonia on the validation dataset.

According to Cochran's Q test, only the performance of Reader 3 significantly changed between the high and low suspicion scenarios, decreasing in the latter (accuracy 70% vs. 78%, $p = 0.008$); no significant changes were found for the other readers ($p > 0.999$). No significant differences in performance were observed between the readers and the R-AI classifier for the high suspicion scenario ($p = 0.369$); on the contrary, a statistically significant

result was obtained for the low suspicion scenario ($p$ = 0.003). However, the post-hoc pairwise McNemar test revealed that the R-AI classifier still had diagnostic performance comparable to that of human readers (lowest $p$ = 0.256), whereas Reader 3 had a significantly lower performance than Reader 2 ($p$ = 0.039) and Reader 4 ($p$ = 0.041). Full statistical results of the comparative analysis are provided in Table 4.

**Table 4.** Comparative analysis of the diagnostic performance of radiomic-based artificial intelligence classifier (R-AI) and human readers in both high and low suspicion scenarios.

| | Cochran's Q Test | | | | | Post-Hoc Pairwise McNemar Test | | | | |
| | High Suspicion Scenario | | Low Suspicion Scenario | | | R-AI | Reader 1 | Reader 2 | Reader 3 | Reader 4 |
| | Accuracy | *p*-Value | Accuracy | *p*-Value | | | | | | |
|---|---|---|---|---|---|---|---|---|---|---|
| R-AI | 79% | | 79% | | R-AI | 1 | - | - | - | - |
| Reader 1 | 74% | | 72% | | Reader 1 | 0.637 | 1 | - | - | - |
| Reader 2 | 79% | 0.369 | 79% | 0.003 * | Reader 2 | 1 | 0.288 | 1 | - | - |
| Reader 3 | 78% | | 70% | | Reader 3 | 0.256 | 1 | 0.039 * | 1 | - |
| Reader 4 | 80% | | 78% | | Reader 4 | 1 | 0.259 | 1 | 0.041 * | 1 |

The *p*-values adjusted after Bonferroni's correction were reported ("*" = statistically significant values).

Finally, considering the subset of 43 CT scans to which two or more radiologists assigned a CO-RADS 3 score, the readers obtained a global accuracy of 55% (95%CI: 47–62%) in the high suspicion scenario and 45% (95%CI: 38–53%) in the low suspicion scenario, whereas the R-AI classifier showed an accuracy of 74% (95%CI: 59–86%). Cochran's Q test was significant in both cases, with $p < 0.001$; however, the post-hoc pairwise McNemar test was significant only for the comparison of the R-AI classifier with Reader 1 ($p$ = 0.023 for both scenarios) and Reader 3 in the low suspicion scenario ($p$ = 0.035). Full details are reported in Tables 5 and 6 and Figure 4.

**Table 5.** Diagnostic performance of radiomic-based artificial intelligence classifier (R-AI) and human readers in classifying the subset ($n$ = 43) of patients who were assigned a CO-RADS 3 score by two or more readers.

| | SE | SP | ACC | PLR | NLR |
|---|---|---|---|---|---|
| | | | High Suspicion Scenario | | |
| Reader 1 | 42% (23–63%) | 35% (14–62%) | 40% (25–56%) | 0.65 (0.37–1.16) | 1.63 (0.79–3.37) |
| Reader 2 | 100% (87–100%) | 6% (0–29%) | 63% (47–77%) | 1.06 (0.94–1.20) | n/a |
| Reader 3 | 42% (23–63%) | 82% (57–96%) | 58% (42–73%) | 2.4 (0.78–7.35) | 0.7 (0.47–1.04) |
| Reader 4 | 92% (75–99%) | 6% (0–29%) | 58% (42–73%) | 0.98 (0.83–1.15) | 1.31 (0.13–13.32) |
| Total | 69% (59–78%) | 32% (22–45%) | 55% (47–62%) | 1.02 (0.83–1.26) | 0.95 (0.61–1.49) |
| | | | Low Suspicion Scenario | | |
| Reader 1 | 12% (2–30%) | 82% (57–96%) | 40% (25–56%) | 0.65 (0.15–2.87) | 1.07 (0.83–1.39) |
| Reader 2 | 31% (14–52%) | 94% (71–100%) | 56% (40–71%) | 5.23 (0.72–38.15) | 0.74 (0.55–0.98) |
| Reader 3 | 0% (0–13%) | 100% (80–100%) | 40% (25–56%) | n/a | 1 (1.00–1.00) |
| Reader 4 | 19% (7–39%) | 88% (64–99%) | 47% (31–62%) | 1.63 (0.36–7.49) | 0.92 (0.71–1.18) |
| Total | 15% (9–24%) | 91% (82–97%) | 45% (38–53%) | 1.74 (0.72–4.23) | 0.93 (0.83–1.04) |
| R-AI | 69% (48–86%) | 82% (57–96%) | 74% (59–86%) | 3.92 (1.36–11.31) | 0.37 (0.20–0.69) |

95% confidence intervals were reported in parentheses. SE: sensitivity; SP: specificity; ACC: accuracy; PLR: positive likelihood ratio; NLR: negative likelihood ratio.

**Table 6.** Comparative analysis of diagnostic performance of radiomic-based artificial intelligence classifier (R-AI) and human readers in classifying the subset (*n* = 43) of patients who were assigned a CO-RADS 3 score by two or more readers.

| Cochran's Q Test | | | Post-Hoc Pairwise McNemar Test | | | | | |
|---|---|---|---|---|---|---|---|---|
| **High Suspicion Scenario** | | | | **R-AI** | **Reader 1** | **Reader 2** | **Reader 3** | **Reader 4** |
| | **Accuracy** | ***p*-value** | | | | | | |
| R-AI | 74% | | R-AI | 1 | - | - | - | - |
| Reader 1 | 40% | | Reader 1 | 0.035 * | 1 | - | - | - |
| Reader 2 | 63% | <0.001 * | Reader 2 | 1 | 0.550 | 1 | - | - |
| Reader 3 | 58% | | Reader 3 | 1 | 0.614 | 1 | 1 | - |
| Reader 4 | 58% | | Reader 4 | 1 | 1 | 1 | 1 | 1 |
| **Low Suspicion Scenario** | | | | **R-AI** | **Reader 1** | **Reader 2** | **Reader 3** | **Reader 4** |
| | **Accuracy** | ***p*-value** | | | | | | |
| R-AI | 74% | | R-AI | 1 | - | - | - | - |
| Reader 1 | 40% | | Reader 1 | 0.023 * | 1 | - | - | - |
| Reader 2 | 56% | <0.001 * | Reader 2 | 0.990 | 1 | 1 | - | - |
| Reader 3 | 40% | | Reader 3 | 0.023 * | 1 | 0.455 | 1 | - |
| Reader 4 | 47% | | Reader 4 | 0.139 | 1 | 1 | 1 | 1 |

The *p*-values adjusted after Bonferroni's correction were reported ("*" = statistically significant values).

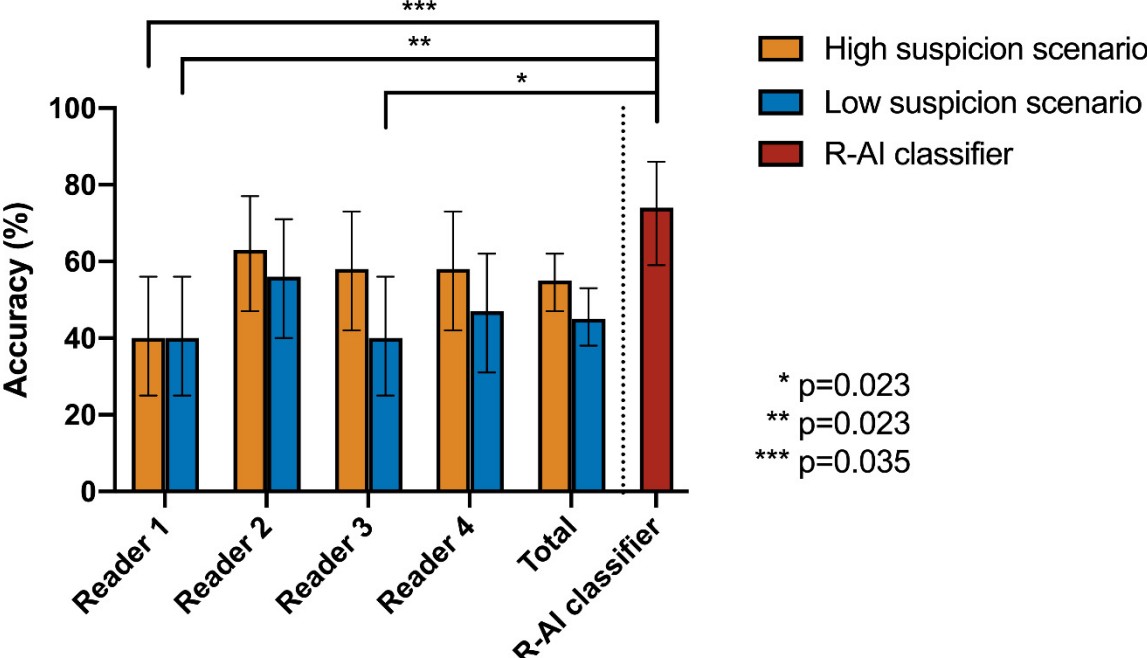

**Figure 4.** Comparison between the performance of the radiomic-based artificial intelligence (R-AI) classifier and the four readers in differentiating COVID-19 from other types of viral pneumonia in the subset (*n* = 43) of patients who were assigned a CO-RADS 3 score by two or more. Bars on graph boxes represent the 95% confidence interval of the accuracy values. Significant results of pairwise McNemar test after Bonferroni's correction were reported.

## 4. Discussion

In this study, the diagnostic performance of multiple readers in distinguishing between COVID-19 and non-COVID-19 pneumonia was evaluated in two different risk scenarios and compared with a radiomic-based artificial intelligence classifier.

Given the well-known complexity of the task, inter-reader agreement in assigning the CO-RADS score was assessed and found to be good, in line with the currently available literature on the reproducibility of this reporting system. Prokop et al. [8] initially observed

an overall Fleiss' kappa of 0.47, but subsequent studies reported a moderate-to-good level of agreement, comparable to that observed in our study [9,26]. Moreover, the absence of significant differences in the diagnostic performance of the three high-experience readers compared to the low-experience reader using the CO-RADS score confirmed the observations by Bellini et al. [10]. On the contrary, in our study, some inconsistency in CO-RADS evaluation was observed for one of the high-experience readers, whose diagnostic accuracy were slightly inferior in the low suspicion scenario.

At the very beginning of the COVID-19 pandemic, a study [7] on 424 patients with COVID-19 and non-COVID-19 viral pneumonia yielded a classification accuracy ranging between 60% and 83% when considering radiologists with direct experience of SARS-CoV-2 infection. Such a wide range of accuracy was reported in subsequent multi-reader analyses [5,27,28], and the results of our study fell within it. The simulation of two different suspicion scenarios allowed us to account for diverse epidemiological conditions, thus providing a more complete picture of the diagnostic performance of the readers.

When applied to the same dataset, the R-AI classifier obtained an accuracy of 79%, comparable to the performance of the human readers in both high and low suspicion scenarios. This result was similar to that reported by Cardobi et al. [29], who developed a radiomic-based model to distinguish COVID-19 from other types of interstitial pneumonia at chest CT. As we used a ten-time larger dataset and applied the R-AI classifier to an independent validation set, our study provided stronger evidence that quantitative imaging and AI models can support this diagnostic task.

Notably, when considering only the subset of patients who were assigned a CO-RADS 3 score by two or more radiologists, the global accuracy of the human readers dropped to 45–55% (depending on the scenario), while the accuracy of the R-AI classifier was almost unchanged (74%). This suggested a more stable performance for the AI, probably based on the extraction of quantitative information within medical images not perceivable by the human brain, even though the result was only partially confirmed by the post-hoc pairwise McNemar test. However, it is reasonable to believe that the smaller sample size, resulting in larger confidential intervals for performance metrics, and correction for multiple comparisons reduced the statistical power by increasing the risk for type II errors. Nevertheless, the result bolsters the concept of AI models helping with equivocal cases, for example, as a second opinion tool to improve diagnostic performance.

AI models with higher performance than our classifier in differentiating between COVID-19 and non-COVID-19 viral pneumonia were also reported, as in the study by Wang et al. [14,30]. However, these authors proposed a method based on single-slice manual segmentation of pulmonary lesions, which is a time-consuming approach hardly feasible in everyday clinical practice compared with our fully automatic approach. Zhou et al. [13] provided another example of an automatic deep learning-based algorithm with very good performance but limited to patients with SARS-CoV-2 and influenza virus infections.

In this regard, contrary to many other similar studies on AI models [16,17], we decided to focus only on the differential diagnosis between COVID-19 and non-COVID-19 viral pneumonia, rather than a broader spectrum of pulmonary diseases. On the one hand, this choice was meant to stress the difficulty posed by the highly overlapping CT findings of these entities; on the other hand, the recognition of typical signs of bacterial infections, such as lobar consolidation, would most likely not require the support of an AI tool. In addition, even if rapid COVID-19 tests are currently widespread and help guide the clinical suspicion, they may be unavailable in some contexts (e.g., night shifts) or provide equivocal results. On the other hand, we envisioned our R-AI classifier as a tool for the radiologist to be used for pneumonia cases whose infectious nature is recognized but with ambiguous or discordant findings compared to clinical history or laboratory results. Nevertheless, in the future, it would be possible to further train the classifier with other lung diseases that mimic COVID-19, such as organizing pneumonia or drug-induced interstitial pneumonia, thus extending its applications.

The main limitation of this study is represented by its retrospective design in a single institution, showing a selection bias. For example, COVID-19 and non-COVID-19 groups had different sample sizes, although this was limited by the fact that the readers were unaware of the case proportions. The R-AI classifier was trained and tested on a COVID-predominant dataset, as well. Additionally, the study population mainly included patients with moderate-to-severe pulmonary involvement based on the visual evaluation of the readers. The underrepresentation of cases with mild disease could represent a bias, even if the sample reflected the actual population for whom chest CT scan is recommended [31]. In addition, the CO-RADS score has been developed specifically for use in patients with moderate to severe disease [8]. Another limitation was that chest CT scans within 15 days from molecular evidence of infection were used, but the cause-and-effect relationship could have been fallacious. Indeed, some of the selected patients may have mixed pneumonia or other diseases. However, the large dataset used should have minimized the impact of this occurrence. Finally, the radiologists were not given clinical information during the evaluation, which could have further improved their performance. In the future, the generalizability of our results should be assessed with a prospective design in a multicenter setting, possibly incorporating clinical information in the AI model.

In conclusion, this work confirmed that distinguishing COVID-19 from other types of viral pneumonia is challenging, even for expert radiologists. Nevertheless, we showed that an artificial intelligence classifier based on radiomic features can provide diagnostic performance in this task comparable to human readers, and probably even better with equivocal cases. Once implemented in the clinical workflow, such a tool could support the radiological activity, for example, by providing a second opinion in case of ambiguous chest CT findings of pulmonary infection.

**Author Contributions:** Conceptualization, F.R., L.B. and L.A.C.; methodology, F.R. and L.B.; software, L.B. and G.Z.; validation, L.B. and G.Z.; formal analysis, F.R., L.B. and G.Z.; investigation, F.R., L.B., G.Z., A.C., F.T., D.A. and L.A.C.; resources, C.V., S.N.M. and F.S.; data curation, F.R., L.B. and G.Z.; writing—original draft preparation, F.R.; writing—review and editing, F.R., L.B. and G.Z.; visualization, F.R., L.B. and G.Z.; supervision, A.V., P.E.C. and A.T.; project administration, A.V., P.E.C. and A.T. All authors have read and agreed to the published version of the manuscript.

**Funding:** This research received no external funding.

**Institutional Review Board Statement:** The study was conducted in accordance with the Declaration of Helsinki, and approved by the Local Ethics Committee of Milan Area C (decision number 188–22042020).

**Informed Consent Statement:** Patient consent was waived because all data were fully anonymized.

**Data Availability Statement:** The datasets generated and/or analyzed during the current study are not publicly accessible but are available from the corresponding authors on reasonable request.

**Conflicts of Interest:** The authors declare no conflict of interest.

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
