# Peer review of "Diagnostic Performance in Differentiating COVID-19 from Other Viral Pneumonias on CT Imaging: Multi-Reader Analysis Compared with an Artificial Intelligence-Based Model"

_tomography, doi:10.3390/tomography8060235_

Round 1
Reviewer 1 Report
The submitted manuscript test the performance of a radiomics-based AI (R-AI) classifier for COVID-19 chest CT from other types of viral-only pneumonia with microbiologically established etiology. The authors used a total sample size of 1031 chest CT scans (n SARS-CoV-2 =647, and n other respiratory viruses =384), where 811 CT scans were used for training and 220 for independent validation. The results obtained from the R-AI classifier were compared to those from 4 reviewers (radiologists). The results were statistically analyzed and showed that R-AI classifier performance was equivalent or superior to human readers (radiologists). The authors concluded that the R-AI classifier might support human readers in the difficult task of distinguishing COVID-19 from other types of viral pneumonia on CT imaging.
Although the manuscript is well written, English editing for grammar and style is needed.
Please simplify the abstract to make it more understandable and attractive for readers.
Please remove the first "figure 1"; you have repetition on figures
Please keep the p values in the table and mention the significant ones using "*."
Thank you
Best Regards
Author Response
Dear reviewer,
Thank you for your valuable suggestions.
- We checked the manuscript to correct grammatical errors and improve linguistic style
- We simplified the abstract to improve its readability
- Redundant “Figure 1” was removed
- Significant p-values in tables are now indicated with “ * “ rather than with bold.
Author Response
Dear reviewer,
Thank you for your valuable suggestions.
- We specified that our AI model belongs to the family of the Artificial Neural Network, employing the Multi-Layer Perceptron architecture.
- We agree that reader 3 showed inconsistency in the CO-RADS evaluation, which is conceivably why the inter-reader agreement was just good (instead of excellent), and the performance of reader 3 was slightly but significantly inferior to the other two readers in the low suspicion scenario. On the contrary, focusing on CO-RADS 3 cases only, the performance dropped for all the readers, with no significant inter-reader differences but demonstrating the superiority of the AI model. Therefore, we believe that reader 3’s inconsistency does not change our conclusions but rather explains the results we found. Nevertheless, we pointed out reader 3’s inconsistency in the Discussion section.